# Cannabinoid Signalling in Immune–Reproductive Crosstalk during Human Pregnancy

**DOI:** 10.3390/biomedicines9030267

**Published:** 2021-03-07

**Authors:** Harmeet Gurm, Jeremy A. Hirota, Sandeep Raha

**Affiliations:** 1Department of Pediatrics and the Graduate Program in Medical Sciences, Faculty of Health Sciences, McMaster University, Hamilton, ON L8N 3Z5, Canada; gurmh@mcmaster.ca; 2Department of Medicine, Faculty of Health Sciences, McMaster University, Hamilton, ON L8N 3Z5, Canada; hirotaja@mcmaster.ca

**Keywords:** endocannabinoid system, inflammation, female reproduction, pregnancy, uterine natural killer cells, extravillous trophoblasts, Delta9-THC

## Abstract

Despite the intricate involvement of the endocannabinoid system in various physiological processes, it remains one of the most under-studied biological systems of the human body. The scope of endocannabinoid signalling is widespread, ranging from modulation of immune responses in innate and adaptive immunity to gestational processes in female physiology. Cannabinoid receptors are ubiquitously distributed in reproductive tissues and are thought to play a role in regulating the immune–reproductive interactions required for successful pregnancy, specifically among uterine natural killer cells and placental extravillous trophoblasts. The use of cannabis during pregnancy, however, can perturb endocannabinoid homeostasis through effects mediated by its major constituents, Δ-9-tetrahydrocannabinol and cannabidiol. Decidualization of the endometrium, invasion, and angiogenesis may be impaired as a consequence, leading to clinical complications such as miscarriage and preeclampsia. In this review, the crosstalk between endocannabinoid signalling in uterine natural killer cells and placental extravillous trophoblasts will be examined in healthy and complicated pregnancies. This lays a foundation for discussing the potential of targeting the endocannabinoid system for therapeutic benefit, particularly with regard to the emerging field of synthetic cannabinoids.

## 1. Introduction

In human pregnancy, the maternal endometrium undergoes a process of decidualization wherein differentiation of endothelial, epithelial and stromal cells, paired with recruitment of immune cells, primes the decidua for blastocyst implantation [1]. This morphological and functional transformation is necessary for communication with placental trophoblasts for the maintenance and success of pregnancy [1]. As fetal-derived trophoblasts begin the process of decidual invasion to remodel the maternal vasculature for utero-placental blood flow establishment, maternal leukocytes come into close contact with trophoblasts, specifically extravillous trophoblasts (EVTs) [2]. This site of decidual immune cell and trophoblast interaction is an immunologically privileged environment since the fetus, a semi-allogenic graft containing both maternal (self) and paternal (non-self) antigens, evades attack and rejection by the maternal immune system [3].

Of the 26–37% of leukocytes localized in the decidua, up to approximately 80% are identified as uterine natural killer (uNK) cells [4,5]. The proportion of uNK cells increases over the menstrual cycle to the first and second trimester decidua basalis, which is the portion of the decidua that underlies the placenta and interacts with EVTs [4,5,6,7]. As pregnancy comes to an end in the third trimester, there is a reduction in uNK cell numbers, suggesting that they have roles in EVT-specific events, such as invasion and vascular remodelling, as opposed to labour and parturition [6]. uNK cells also contribute to the maintenance of pregnancy through the production of cytokines and angiogenic growth factors [8,9,10,11]. Other decidual immune cells include macrophages and T cells, however research surrounding their functions in pregnancy is limited [4]. Given that uNK cells are the predominant decidual leukocyte population and have established functions in pregnancy, immune responses in female reproduction mediated by uNK cells are the focus of this review.

The endocannabinoid system (ECS) plays a necessary role in a variety of physiological processes in the human body, including modulation of immune responses and reproductive events [12]. Unfortunately, research surrounding ECS regulation in peripheral tissues is limited, and has primarily focused on the central nervous system (CNS) wherein the ECS is implicated in assisting with pain perception and cognitive functions [12]. The effects mediated by the ECS are exerted through endogenous cannabinoids (endocannabinoids), the enzymes involved in endocannabinoid biosynthesis and degradation, and cannabinoid receptors [13]. Anandamide (AEA) was the first identified endocannabinoid in 1992, after which its main biosynthetic and hydrolytic enzymes were determined to be *N*-acyl-phosphatidylethanolamine-specific phospholipase D (NAPE-PLD) and fatty acid amide hydrolase (FAAH), respectively [14,15,16,17]. Shortly after, the second major endocannabinoid was identified as 2-arachidonoyl glycerol (2-AG) [18]. 2-AG is synthesized by diacylglycerol lipase (DGL) and degraded by monoacylglycerol lipase (MAGL) [19,20,21,22]. The targets of these endocannabinoids are cannabinoid receptors 1 (CB1R) and 2 (CB2R), which are the primary receptors of the ECS [13,23]. CB1R is most abundant in the CNS, although it is also expressed in the peripheral nervous system and peripheral cell types [24,25]. In contrast, CB2R is predominantly localized in cells of the immune and reproductive systems [26,27]. Other receptors of the ECS include the transient receptor potential (TRP) ion channels, peroxisome proliferator activated receptors (PPARs), G protein-coupled receptor 55 (GPR55), GPR18, and GPR119 [28,29,30,31,32]. Unlike CB1R and CB2R, the roles of these non-classical receptors are poorly characterized.

Both CB1R and CB2R are classified as GPCRs that regulate an array of signal transduction cascades, particularly the adenylyl cyclase pathway [33]. Cannabinoid receptor activation is associated with attenuated adenylyl cyclase through guanine nucleotide-binding protein complex (G_i_)-mediated actions [34,35]. Cyclic adenosine monophosphate (cAMP) levels and protein kinase A (PKA) activity are subsequently inhibited [36,37,38]. The functional responses impacted by cannabinoid-dependent inhibition of adenylyl cyclase include cellular apoptosis, differentiation, proliferation, and survival [39,40,41,42].

The presence of ECS components at the maternal-fetal interface dates to 1999 when CB1R and CB2R localization was demonstrated in the human placenta and the BeWo cell line, which is representative of cytotrophoblasts [43]. Production of NAPE-PLD, FAAH, MAGL, and DGL has also been identified in reproductive tissues [44,45,46]. In comparison, the presence of endocannabinoid signalling in uNK cells has been recently reported [47]. The evidence for CB1R, CB2R, NAPE-PLD, and FAAH expression in uNK cells suggests that this decidual leukocyte population may have a role in regulating local levels of AEA via its regulatory enzymes [47].

Furthermore, alterations in ECS signalling during early pregnancy are associated with impaired decidualization due to inhibited differentiation of endometrial stromal cells, and decreased secretion of cytokines and angiogenic growth factors by uNK cells [48,49]. These effects may then interfere with EVT-specific functions of invasion and angiogenesis [45,50]. One of the ways that ECS signalling in female reproduction can be disrupted is through exposure to plant-derived cannabinoids (phytocannabinoids), such as delta-9-tetrahydrocannabinol (∆^9^-THC) and cannabidiol (CBD) [45,51]. In fact, compromised decidualization is associated with various adverse clinical outcomes, including implantation failure, recurrent spontaneous miscarriage, preeclampsia, fetal intrauterine growth restriction, and preterm birth [49]. This review will focus on the implications of ECS-mediated crosstalk between human (uNK) cells and EVTs at the maternal-fetal interface in successful pregnancies and those burdened with complications.

## 2. Uterine Natural Killer Cells in the Decidua

uNK cells are a phenotypically distinct subset of human natural killer (NK) cells as they are cluster differentiation 56 positive (CD56^+)^, CD2^+^, CD3^−^ and CD19^−^ [4,52]. The CD56 (NKH-1/Leu-19) antigen is a neural cell adhesion molecule expressed mainly on NK cells, and is involved in embryogenesis [5,53]. About 90% of peripheral NK cells have low-density CD56 antigenic expression and are characterized as CD56^dim^ while the remaining, including uNK cells, are CD56^bright^ [54]. However, unlike peripheral CD56^bright^ NK cells, uNK cells possess lytic granules that contain perforin and granzyme [55,56]. uNK cells are also functionally different as they do not express the CD19 antigen, which mediates antibody-dependent cellular cytotoxicity [57]. As a result, the cytotoxicity of uNK cells is approximately 30% less than that of peripheral CD16^+^ NK cells [58].

Besides differences in cytotoxic functionality, a microarray comparison of CD56^bright^ uNK cells with CD56^bright^ and CD56^dim^ peripheral NK cells determined that 278 genes out of the 10,000 analyzed are differentially expressed by three-fold [56]. The majority of these genes were overexpressed in uNK cells, including those involved in regulating cytotoxicity such as the NK cell C-type lectin-like receptors (NKG2C, NKG2E) and killer cell immunoglobulin-like receptors (KIR3DL1, KIR3DL2, KIR2DL3, KIR2DL4) [56]. Expression of genes that modulate immune cell functions, such as galectin-1 and placental protein-14, was also elevated. In addition, the cell-surface molecules CD9, CD151, and tetraspan-5 were reported to be exclusively expressed in uNK cells [56].

The main function of uNK cells appears to be the secretion of cytokines, growth factors, and angiogenic factors to assist in trophoblast invasion and maternal spiral artery remodelling [54,59]. uNK cells isolated from early pregnancy deciduae show gene expression and secretion of the cytokines tumour necrosis factor-alpha (TNF-α), interferon-gamma (IFN-γ), leukemia inhibitory factor (LIF), interleukin-1beta (IL-1β), IL-10, and transforming growth factor-beta (TGF-β) [8,9,10]. The growth factors granulocyte colony-stimulating factor (G-CSF) and granulocyte macrophage colony-stimulating factor (GM-CSF) are also produced by uNK cells [8]. Moreover, uNK cells are a source of angiogenic factors, such as angiopoietin-1 (Ang-1), angiopoietin-2 (Ang-2), and vascular endothelial growth factor (VEGF) [11].

uNK cells have also been implicated in gestational complications. Endometrial biopsies obtained from women suffering from recurrent miscarriage show an increase in CD56^+^ NK cells [60,61]. However, these investigations have not considered the intensity of CD56 antigenic expression, which makes it unclear whether elevated CD56^bright^ uNK cells or CD56^dim^/CD56^bright^ peripheral NK cells are implicated [60]. Conversely, no correlation between uNK cell number and recurrent miscarriage has been observed as well [62]. Placental bed biopsies have further displayed elevated CD56^bright^ NK cell numbers in pregnancies complicated with preeclampsia and fetal growth restriction [63,64]. This is linked to increased secretion of IL-12 and IFN-γ by NK and T helper 1 (Th1) cells [64]. Both IL-12 and IFN-γ play roles in promoting an inflammatory response, which is a hallmark of preeclampsia [64]. Further investigations examining the functions of human uNK cells are required to elucidate how decidual immune responses may contribute to pathological reproductive events.

## 3. Trophoblasts in the Placenta

In addition to decidualization, successful pregnancy depends on proper trophoblast differentiation and functionality [65,66]. In brief, human placental development begins with the formation of the trophectoderm around 5 days post-fertilization [65,66]. This separates the trophectoderm, the lineage that interacts with the endometrium, from the inner cell mass that gives rise to the embryo and fetus [65,66]. As pregnancy progresses to implantation, mononuclear cytotrophoblasts (CTBs) are derived from trophectoderm stem cells [65,66]. Approximately 2 weeks post-fertilization, CTBs begin to proliferate, differentiate, and undergo cell–cell fusion to become multi-nucleated syncytiotrophoblasts (STBs) [65,66]. STBs form the syncytium, which establishes the primary site of maternal-fetal exchange of gases and nutrients [65,66]. STBs are also major producers of pregnancy-specific hormones, such as human chorionic gonadotropin [65,66].

Around 3 weeks post-fertilization, CTBs differentiate into another type of trophoblast, the EVTs [65,66]. EVTs are primarily responsible for invasion into the maternal decidua and migration into the spiral arteries to establish low resistance blood flow to the uterus [65,66]. CTBs therefore give rise to 2 major trophoblast lineages in the placenta, with each possessing distinct roles during placentation and pregnancy [65,66]. Of these, only EVTs come into close contact with leukocyte populations in the decidua basalis [7]. Since uNK cells are the largest decidual immune cell population, communication between EVTs and uNK cells is the focus below.

## 4. Interactions between uNK Cells and EVTs

Peripheral NK cells display cytotoxic activity against infected and foreign cells, particularly those that have reduced expression of human leukocyte antigen class I (HLA-I) molecules [67]. This response is dependent on the balance of activating and inhibitory receptors present on NK cells [67]. However, uNK cells do not show cytotoxic activity against EVTs despite their close proximity in the decidua and presence of foreign paternal antigens [68]. EVTs are the only trophoblast type with expression of HLA-I molecules, including the classical HLA-C and non-classical HLA-E and HLA-G [69,70]. Due to the presence of HLA-I molecules on EVTs, inhibitory receptor interactions allow EVTs to evade attack by uNK cells [71]. HLA-C is a ligand for KIR2D receptors, which can either activate or inhibit uNK cell cytolytic activity against EVTs depending on the specific type [72,73]. KIR2DL1, KIR2DL2 and KIR2DL3 receptors inhibit while KIR2DS1 and KIR2DS2 activate uNK cell responses [67]. Moreover, HLA-E is recognized by CD94/NKG2 receptors, and HLA-G is recognized by KIR2DL4 [72,74]. Interactions between HLA-I molecules and inhibitory NK cell receptors are also associated with promoting trophoblast invasion and establishing fetal circulation [73,74].

The increase in number of uNK cells in the late secretory phase of the menstrual cycle and early pregnancy can be attributed to increased proliferation, differentiation of peripheral CD56^dim^ NK cells to uNK cells, and migration of CD56^bright^ peripheral NK cells [75,76,77]. EVTs are implicated in the latter mechanism as they may assist NK cell movement into the decidua [77]. Specifically, interactions between chemokine receptor type 3 (CXCR3) and its ligand CXCL9, as well as CXCR4 and CXCL12, are important for peripheral NK cell migration [77]. CD56^bright^ peripheral NK show increased expression of CXCR3 and CXCR4 compared to CD56^dim^ peripheral NK cells [77]. Immunohistochemical staining has demonstrated that invading EVTs also express CXCL12 [77]. Due to the expression of CXCR4 on NK cells and CXCL12 on EVTs, trophoblasts are suggested to play a role in attracting CD56^bright^ peripheral NK cells to the decidua through CXCR4-CXCL12 interactions in early pregnancy [77].

Communication between uNK cells and EVTs is also important for the regulation of trophoblast invasion [78]. uNK cells isolated and purified from the decidua basalis of first trimester pregnancies are major producers of genes responsible for encoding chemokines, including IL-8 (CXCL8) and interferon-inducible protein-10 (IP-10) [78]. Isolated EVTs are shown to express CXCR1 and CXCR3, which are receptors that bind to IL-8 and IP-10, respectively [78]. Furthermore, in vitro Transwell migration assays with different NK cell subtypes have demonstrated that only uNK cells promote migration and invasion of trophoblasts [78]. uNK cells secrete elevated levels of IL-8 and IP-10, resulting in greater CXCR1- and CXCR3-mediated interactions that play a role in inducing trophoblast cell migration and invasion [78].

In addition, matrix metalloproteinases (MMPs) are necessary for the breakdown of the decidual extracellular matrix, which is an event required for invasion and angiogenesis [79]. The activity of MMPs is tightly regulated by tissue inhibitors of metalloproteinases (TIMPs) [80]. Through immunohistochemical staining, EVTs have been shown to express MMP-1, MMP-2, MMP-3, MMP-9, and their inhibitors, TIMP-1 and TIMP-2 [80]. uNK cells also express and secrete MMP-2, MMP-9, TIMP-1, and TIMP-2 [81]. Moreover, EVTs cultured in uNK cell supernatant from weeks 12–14 of gestation demonstrate enhanced invasion through a mechanism involving increased secretion of MMP-9 and decreased expression of M30, a biomarker for apoptosis [82].

Alongside facilitating trophoblast invasion, first trimester uNK cells possess important roles in vascular remodelling as they secrete VEGF, placental growth factor (PlGF), Ang-1, and Ang-2 [11,78,83]. EVTs are also a major source of VEGF, PlGF, Ang-1, Ang-2 as well as angiogenin [84]. All of these factors are positive regulators of angiogenesis, with VEGF serving as the key mediator in embryogenesis [85]. During pregnancy, the process of vascular remodelling follows 4 distinct stages that are dependent on the extent of vascular smooth muscle cell and endothelial cell disruption, and EVT colonization [86]. EVT presence is generally increased as remodelling progresses [86]. However, the number of leukocytes, particularly uNK cells, is shown to increase prior to EVT infiltration [86]. Remodelling in the decidua thus begins before invasion by trophoblasts, and is supported by uNK cells through the production of angiogenic factors [86,87]. The number of uNK cells also decreases around week 20 of gestation, the time when angiogenesis is complete in the decidua [81,86]. Therefore, the major functions of uNK cells involve assisting in the establishment and maintenance of pregnancy through crosstalk with EVTs during invasion and vascular remodelling.

## 5. ECS Signalling in the Immune System

The immune system functions to defend the body from foreign substances and infection through two lines of defense: innate and adaptive responses [88]. The expression of cannabinoid receptors, CB1R and CB2R, is reported in the major cells involved in innate (dendritic cells, monocytes/macrophages, neutrophils, NK cells) and adaptive (B cells, T cells) immunity [89,90,91]. Of these, B cells express the highest levels of CB1R and CB2R, followed by NK cells, monocytes/macrophages, and lastly, T cells [90]. CB1R and CB2R are also expressed in lymphoid tissues, such as the spleen and thymus [89,90,91]. Despite the presence of both cannabinoid receptors, the gene expression of CB2R is 10- to 100-fold greater than that of CB1R in immune cell populations [90]. In addition, lipopolysaccharide (LPS)-stimulated macrophages, dendritic cells, and T cells have been shown to synthesize the endocannabinoids AEA and 2-AG, which is suggestive of endocannabinoid level regulation by immune cells [91,92,93].

Cannabinoid receptor activation in cell types of the innate immune system is associated with apoptosis and recruitment of dendritic cells, inhibited migration of neutrophils, inhibited phagocytic activity and Th1-type cytokine secretion of macrophages, and inhibited cytotoxicity of NK cells [94,95,96,97,98,99]. Endogenous and exogenous cannabinoids are also reported to modulate local inflammatory responses through interactions with toll-like receptors (TLRs) present on macrophages and dendritic cells [100]. TLRs are pattern-recognition receptors (PRRs) that identify pathogen-associated molecular patterns (PAMPs) to initiate innate immune responses [100]. TLR2 and TLR4 signalling can be suppressed by cannabinoids through downstream inhibition of the activity of transcription factor nuclear factor-ᴋB (NF-ᴋB) via the myeloid differentiation factor 88 (MyD88) pathway [100]. In particular, activation of CB2R by its synthetic agonist JWH 133 (1.5 mg/kg) is associated with suppressed TLR4 signalling in dendritic cells of B10.RIII and BALB/c mice, resulting in attenuated cellular activation and maturation [101]. The phytocannabinoid ∆9-THC (5–20 µM) has also been shown to inhibit inducible nitric oxide transcription and nitric oxide production in murine RAW 264.7 macrophages through a CB2R-dependent mechanism [102].

ECS signalling modulates events in B cells and T cells of the adaptive immune system as well. The majority of experimental investigations, however, have examined B cell functions in mouse models. In BALB/c mice, 2-AG (1 µM) stimulated naïve B cell chemotaxis through a CB2R-dependent mechanism [103]. CB2R also mediated immature B cell retention in the bone marrow sinusoids of chimeric C57BL/6 (Ly5.2^+^) and Boy/J (Ly5.1^+^) mice, which is linked to reduced maturation [104]. Moreover, activation of CB2R is associated with immunoglobulin class switching from IgM to IgE in B cells of C57BL/6 mice [105]. On the other hand, investigations with primary human CD3^+^ T cells demonstrate that AEA (2.5 µM) suppressed the proliferation and secretion of TNF-α, IFN-γ, and IL-2 [106]. AEA (40 nM) has also been implicated in the attenuated migration and pro-inflammatory IL-17 secretion observed in primary human CD8^+^ T cells and CD4^+^ Th17 cells, respectively [106,107]. In addition, ∆^9^-THC (5 µM) is reported to balance CD3^+^ Th-cell responses by decreasing the ratio of Th1/Th2 cytokines, therefore providing this phytocannabinoid with an immunomodulatory role [108]. Similar to B cells, cannabinoid-dependent outcomes in T cells are primarily mediated through CB2R signalling [106,107,108].

## 6. ECS Signalling in the Female Reproductive System

In healthy human pregnancy, plasma levels of AEA are tightly regulated from ovulation to the onset of labour [109]. Beginning with ovulation, an increase in AEA is favoured, suggesting a potential role for this endocannabinoid in follicular maturation [110]. For the majority of pregnancy, however, low AEA levels are maintained for fertilization, implantation, decidualization, and placentation [109,111]. As pregnancy comes to an end, levels of AEA then experience a surge to assist in labour and parturition [109,111]. In fact, a cross-sectional investigation following healthy singleton pregnancies in humans reported mean AEA levels of 0.89, 0.44 and 0.42 nM in the first, second and third trimesters, respectively [111]. These levels increased 6-fold in labouring women, reaching a plasma concentration of 2.5 nM [111]. Therefore, relatively low and stable AEA levels contribute to the success of pregnancy as imbalances are linked to gestational complications, such as miscarriage, preeclampsia and ectopic pregnancy [47,111].

In comparison, the levels and roles of 2-AG in human pregnancy remain undefined, although its regulation over pregnancy in mice has been explored. In pregnant CD-1 mice, 2-AG (1–10 nM) arrested blastocyst development in a dose-dependent manner, indicating that elevated 2-AG is detrimental during the pre-implantation period [112]. While outcomes from mouse models may be translated to humans, it is important to consider that gestational length varies across species, lasting 3 weeks in mice and 40 weeks in humans [113]. As a result of differences in physiology, there are changes in the processes of decidualization, placentation, and development of the maternal-fetal interface between mice and humans [113]. Finally, expression of both CB1R and CB2R has been reported in the human uterus, decidua, placenta and trophoblast cells [44,47,114,115].

Another important role for endocannabinoid signalling during pregnancy is the modulation of trophoblast functions [116]. In the human choriocarcinoma-derived BeWo cells, AEA (10 μM) attenuated cellular proliferation [44]. 2-AG (10–25 μM) also inhibited proliferation in primary human CTBs as well as induced apoptosis through chromatin condensation and fragmentation, presence of apoptotic bodies, and generation of reactive oxygen and nitrogen species [117]. Furthermore, the cannabinoids ∆^9^-THC (20 µM), AEA (10 µM), and 2-AG (10 µM) have been shown to impair the biochemical and morphological differentiation of CTBs, leading to poor trophoblast cell-to-cell fusion [50,118,119]. ∆^9^-THC (20 µM) exposure also decreased the secretion of human chorionic gonadotropin, placental lactogen, and insulin-like growth factors in BeWo cells, all of which are important for fetal development and pregnancy success [50]. Finally, ∆^9^-THC (10 mM) has been associated with decreased trophoblast invasion in the human HTR-8/SVneo cell line that models EVTs [120].

## 7. Crosstalk between ECS Signalling in the Immune and Reproductive Systems

One of the major immune-reproductive interactions in human physiology occurs between decidual uNK cells and placental EVTs (Figure 1) [3]. In these systems and cell types, the ECS contributes to the maintenance of pregnancy. Despite this central role, connections between ECS signalling in the immune and reproductive systems during pregnancy have not been clearly elucidated.

Decidualization is an early pregnancy-specific event wherein the endometrium is transformed into the decidua through the differentiation of endometrial stromal cells, and the migration of peripheral leukocytes [121]. Increased cAMP production plays a critical role during the decidualization process, which perhaps underlies the importance of cannabinoid signalling due to its use of this second messenger [121]. Specifically, elevated levels of cAMP are associated with transcription of decidual prolactin (PRL) in differentiated endometrial stromal cells around day 25 of the menstrual cycle to prepare for blastocyst implantation and pregnancy establishment [123]. uNK cells also contribute to decidualization by secreting elevated levels of PRL when stimulated by cAMP (Figure 1) [124]. However, cannabinoid receptor signalling is associated with inhibited cAMP activity, which can then lead to disrupted decidualization [121,122]. In the absence of cAMP, endometrial stromal cells no longer differentiate and cease to express decidualization markers PRL and insulin-like growth factor binding protein-1 (IGFBP-1) [123,125]. A similar outcome may be observed in uNK cells in response to cannabinoid receptor activation.

The role of the ECS in regulating decidual function has been investigated using the human endometrial stromal cell line, St-T1b. Treatment of St-T1b cells with AEA (10 µM) is associated with cell cycle arrest at the G_2_/M phase in undifferentiated cells and attenuated proliferation in differentiated cells [49]. AEA exposure also decreased the expression of PRL and IGFBP-1 [49]. Since St-T1b cells express CB1R, but not CB2R, the effects of AEA on attenuated decidualization were initially postulated to occur through a CB1R-dependent mechanism [49]. Further investigations confirmed that activation of CB1R by a synthetic agonist, WIN-55,212-2 (2–10 µM), decreased expression of PRL and IGFBP-1 in endometrial stromal cells through a cAMP-dependent mechanism [122]. This outcome was reversed by treatment with the synthetic CB1R antagonist AM251 (1 µM), which showed further evidence that CB1R activation negatively modulates decidualization [122]. The ECS is thus implicated in impaired decidualization through CB1R signalling, which inhibits adenylate cyclase activity and cAMP levels required for decidual cell differentiation [122].

In human uNK cells isolated and purified from first trimester decidual tissues, alterations in the balance between AEA synthesis and degradation have been linked to impaired decidualization [48]. Relative to elective pregnancy terminations, miscarried samples were characterized by a two-fold increase in AEA levels [48]. While no differences in CB1R and CB2R expression were observed, uNK cells from women who experienced miscarriage showed elevated levels of both NAPE-PLD and FAAH, suggesting that local production of AEA by NAPE-PLD increased to a greater degree than its breakdown by FAAH [48]. uNK cells from miscarried deciduae also displayed reduced expression of PRL and IGFBP-1 [48]. However, there is a lack of research investigating CB1R-mediated signalling and its effects on cAMP regulation in uNK cells compared to endometrial stromal cells.

The process of EVT invasion is dependent on successful decidualization as trophoblasts come into close contact with decidual leukocytes for the remodelling of maternal vasculature [78]. Complications during decidualization can therefore compromise the crosstalk between uNK cells and EVTs. In particular, both uNK cells and EVTs produce common factors, including TNF-α and IL-25, that are implicated in early pregnancy [126,127,128]. TNF-α is an immune-activating Th1 cytokine with roles in implantation, and the regulation of trophoblast functions such as apoptosis, differentiation, invasion, and syncytialization [129,130,131,132]. In EVTs isolated from first-trimester placentae following elective terminations, TNF-α (10 ng/mL) treatment inhibited cellular migration and invasion [132]. TNF-α also appears to modulate decidualization as it suppressed PRL production in primary endometrial stromal cells [133]. Primary uNK cells demonstrate decreased levels of both PRL and IGFBP-1 after TNF-α treatment as well [48]. Similar results are reported in women who have suffered from miscarriage as their uNK cells secrete elevated TNF-α [48]. Other gestational conditions associated with high TNF-α levels include preterm birth and preeclampsia [134,135].

Moreover, IL-25 (IL-17E) is involved in mediating Th2-type immune responses [127]. Treatment with IL-25 (1 ng/mL) has been shown to stimulate proliferation and decidualization through elevated levels of PRL and IGFBP-1 in endometrial stromal cells [124,127]. While little is known about the effects of IL-25 in EVTs and uNK cells, this cytokine is downregulated in women who have experienced recurrent miscarriage [136]. Since IL-25 is implicated in the success of decidualization, its production by EVTs and uNK cells may contribute to subsequent events in pregnancy, such as invasion and angiogenesis, through modulation of the balance between Th1 and Th2 cytokines [127].

To conclude, alterations in cannabinoid receptor-mediated signalling can impair decidualization, which can subsequently interfere with EVT-uNK cell crosstalk as well as lead to compromised trophoblast invasion and vasculature remodelling (Figure 1). Complications during pregnancy may occur as a consequence, including failed implantation, intrauterine growth restriction, spontaneous recurrent miscarriage, preeclampsia, and preterm birth [49].

## 8. Effects of Maternal Cannabis Use during Pregnancy

Cannabis is one of the most commonly used drugs during pregnancy, with a self-reported incidence of 1.4–7.1% [137,138,139]. Among these pregnant women, the majority revealed cannabis use in the first trimester during their prenatal clinic visits [140]. This increased consumption in early pregnancy may be attributed to the perceived effects of diminished nausea and vomiting [140]. Despite the anecdotal benefits, conflicting clinical data suggests that chronic cannabis use is linked to cannabinoid hyperemesis syndrome (CHS) [141]. CHS is characterized by episodes of acute nausea, vomiting, and abdominal pain persisting for 24–48 h post-cannabis use [141]. As of 2020, only 6 cases of CHS have been published with pregnant patients [142]. This may be the result of individuals not disclosing cannabis use to health care providers when patient history is recorded due to perceived social and medical stigma [142].

Maternal cannabis use is also associated with various complications for both the mother and fetus, including decreased fetal birth weight (<2500 g), increased admissions to the neonatal intensive care unit (NICU), preterm birth (<37 weeks of gestation), small-for-gestational-age (SGA) status, miscarriage, and placental abruption [143,144]. Moreover, prenatally exposed children display lower scores in verbal reasoning, language comprehension, reading tasks, memory, and visual function, compared to non-exposed children during infancy and adolescence [144]. Children also show deficits in impulse control and tasks requiring sustained attention [144]. Since research examining the mechanistic effects of cannabis use during pregnancy and on fetal development is limited, there is a requirement to develop clear safety data to educate patients who may not be aware of the potential adverse effects associated with its consumption [144,145]. This knowledge may further assist in developing clinical interventions to mitigate the risk of gestational complications and ensuing health outcomes for children exposed to cannabis in utero.

In addition, the primary bioactive phytocannabinoids, ∆^9^-THC and CBD, can readily cross the placenta due to their lipophilic nature [143]. Human investigations of Δ^9^-THC transfer across the placental barrier have not been evaluated, however animal models indicate that 10–28% of its maternal plasma concentrations are found in the fetal circulation [146,147]. While similar outcomes may be observed in humans, transfer of ∆^9^-THC is dependent on route of administration, placental permeability, and individual variations in frequency of use [143,148]. Unfortunately, relatively little is known about the effects of CBD during human pregnancy.

In terms of pharmacological activity, ∆^9^-THC is a partial agonist for CB1R (Ki = 5.05 nM) and CB2R (Ki = 3.13 nM) (Figure 2) [149,150]. CBD has a more complex mechanism of action as it is a non-competitive allosteric modulator of CB1R (Ki = 4350 nM) and partial agonist of CB2R (Ki = 2860 nM) [150,151]. In comparison, the endocannabinoid AEA is a partial agonist for CB1R (Ki = 209 nM) and is largely inactive at CB2R (Ki = 1940 nM) whereas 2-AG is a full agonist for both CB1R (Ki = 34.6 nM) and CB2R (Ki = 145 nM) [25,152,153,154]. Besides differences in cannabinoid receptor affinity, only endocannabinoids possess the ability to initiate a negative feedback mechanism via retrograde signalling to suppress further neurotransmitter release [155]. Since phytocannabinoids lack this regulatory mechanism, exposure to cannabis can perturb ECS signalling through persistent cannabinoid receptor activation and stimulation [155].

Another point of contrast is that endocannabinoids are associated with local effects whereas phytocannabinoids act systemically and can influence interactions across systems, including those between uNK cells and EVTs in pregnancy [156]. Treatment with ∆^9^-THC (40 µM) in chorionic villous explant cultures has been associated with increased local AEA levels after a 72-h exposure period [45]. Since ∆^9^-THC can mediate its actions through CB1R and has been shown to alter ECS homeostasis in the placenta, it may interfere with placental-decidual communication. Specifically, ∆^9^-THC exposure may compromise the crosstalk between uNK cells and EVTs by altering their production of cytokines and growth factors, which are required for successful decidualization, invasion, and angiogenesis.

## 9. Therapeutic Applicability of Cannabinoids

Due to its multi-faceted biological roles, the ECS holds therapeutic promise for conditions associated with altered endocannabinoid homeostasis [157]. However, targeting components of the ECS for drug design is associated with challenges due to non-specific responses that result from global cannabinoid receptor activation [157]. In the 2-year rimonabant in obesity (RIO)-North America clinical trial, efficacy and safety data surrounding rimonabant, a selective CB1R inverse agonist (Ki = 1.8 nM), was investigated [158,159]. The primary outcomes in overweight and obese participants were decreased body weight and waist circumference by up to 48.6% in the first year, with maintained weight loss in the second year [158]. Unfortunately, 83–86% of participants receiving rimonabant also reported severe psychological side-effects, such as depression and anxiety [155]. Rimonabant was not approved by the U.S. Food and Drug Administration (FDA) or Health Canada [158,160]. A lesson learned from rimonabant is the requirement for biased ligand signalling in cannabinoid-based therapeutics to create targeted, site-specific pharmacological responses [161].

As of early 2021, the FDA has approved 4 cannabinoid-based drugs: Epidiolex, Cesamet, Marinol, Syndros [162,163]. Cesamet and Marinol are also approved by Health Canada, alongside Sativex [164]. Of these, only Epidiolex and Sativex contain phytocannabinoids whereas the rest are derived from manufactured compounds classified as synthetic cannabinoids [162,164,165].

## 10. Synthetic Cannabinoid-Based Drugs: Cesamet, Marinol, Syndros

Nabilone is a synthetic analogue of ∆^9^-THC with the trade name Cesamet [166]. The approved indication for nabilone is cancer chemotherapy-induced nausea and vomiting (CINV) [164]. While the mechanisms behind CINV are complex, it involves the stimulation of the dorsal vagal complex in the brainstem, following which the chemoreceptor trigger zone allows compounds to cross the blood–brain barrier to induce emesis [167,168]. The dorsal vagal complex is the central emetic structure and contains only CB1R, however both CB1R and CB2R are present in peripheral emetic structures such as the enteric nervous system [168]. The ability of nabilone, and other cannabinoids as well, to serve as anti-emetic agents has been shown to be mediated primarily through interactions with CB1R [169]. In terms of pharmacokinetics, nabilone is a high affinity agonist for CB1R (Ki = 2.89 nM) and CB2R (Ki = 1.84 nM) [166,170]. It also reaches peak plasma concentrations 2–4 h after administration, and has a half-life of about 2 h [171]. In contrast, the pharmacological profile of ∆^9^-THC varies depending on route of administration, although it has been shown to reach peak plasma concentrations within minutes of inhalation, and has a half-life of 21.5 h [172].

Dronabinol is another cannabinoid that contains synthetically manufactured ∆^9^-THC [173]. This compound is marketed under the brand name Marinol as a capsule, and Syndros as an oral solution [160,168]. Dronabinol is approved for CINV, and acts via a mechanism of action similar to nabilone to alleviate nausea and vomiting [169]. It is also indicated for anorexia associated with weight loss in acquired immunodeficiency syndrome [169]. Cannabinoids, such as ∆^9^-THC, can stimulate appetite through activation of CB1R in areas of the CNS that regulate energy balance, specifically the hypothalamus and brainstem [174,175,176]. CB1R is also localized in mesolimbic system, the area of the brain associated with reward-related feeding, and in metabolically active peripheral organs [176]. Since dronabinol is a synthetic form of ∆^9^-THC, it is similarly an agonist for both CB1R and CB2R [149,150]. Dronabinol reaches peak plasma concentrations after approximately 1–1.5 h, and has a half-life of 2–6 h depending on its formulation (capsule: 2–4 h, oral solution: 5–6 h) [173].

## 11. Phytocannabinoid-Containing Drugs: Epidiolex, Sativex

Epidiolex is an oral solution of CBD that is used for treating seizures in the Lennox-Gestaut and Dravet syndromes [177]. Although CBD’s mechanism of action in ameliorating epileptic symptoms in humans remains unknown, one of its proposed pathways involves the TRP ion channels [177,178]. The expression of transient receptor potential cation channel subfamily V member I (TRPV1) is increased in epilepsy, and linked to elevated neuronal hyperexcitability [179,180]. In rat hippocampal brain slices perfused with artificial cerebrospinal fluid, which serves as an in vitro model of epileptiform activity, CBD has been shown to activate and then rapidly desensitize TRPV1 [179]. This is associated with downregulated neuronal excitability and may be a potential mechanism linked to CBD’s anti-convulsant effects [179].

Finally, nabiximols is marketed under the trade name Sativex as an oromucosal spray containing ∆^9^-THC and CBD in equimolar amounts [181]. It is utilized as an adjunctive treatment for spasticity and neuropathic pain associated with multiple sclerosis [181]. Similar to Epidiolex, the mechanism of action for nabiximols is poorly understood. However, nabiximols has been shown to inhibit cortical and spinal excitability, which is then associated with improved spasticity [182]. These effects are suggested to occur through the modulation of inhibitory γ-aminobutyric acid and excitatory glutamate neurotransmitter release following CB1R activation [183,184]. While nabiximols is characterized as a partial agonist for CB1R and CB2R, an analysis of its inhibitory constant values is not yet available [182].

## 12. Conclusions

Human pregnancy presents a unique dilemma wherein the maternal immune system must accept the fetus, which is comparable to a semi-allogenic graft, and create a microenvironment conducive to its growth. The interactions between uNK cells and EVTs are necessary for this process, particularly since these cell types work together for successful decidual invasion and remodelling of the maternal vasculature. Furthermore, components of the ECS are involved in regulating diverse biological processes, including immune and reproductive events in pregnancy. In cases of maternal cannabis use, however, endocannabinoid homeostasis and signalling may become dysregulated. An important consideration is that phytocannabinoids lack the retrograde feedback mechanism utilized by endocannabinoids and can thus continuously activate the cannabinoid receptors. In particular, activation of CB1R by phytocannabinoids is associated with inhibited decidualization through a cAMP-dependent mechanism. As a consequence of cannabis exposure, the crosstalk between uNK cells and EVTs can become compromised, resulting in impaired invasion, angiogenesis, and potentially, later pregnancy events that promote fetal development.

In addition, while some synthetic cannabinoids are approved by the U.S. FDA and Health Canada, other formulations are unregulated and lack data pertaining to drug interactions and long-term outcomes. If consumed by pregnant women, synthetic cannabinoids may have consequences for pregnancy establishment, maintenance, and success. Therefore, further molecular and clinical research is necessary to ascertain how endocannabinoids, phytocannabinoids, and synthetic cannabinoids may impact immune–reproductive interactions during pregnancy.

## Figures and Tables

**Figure 1 biomedicines-09-00267-f001:**
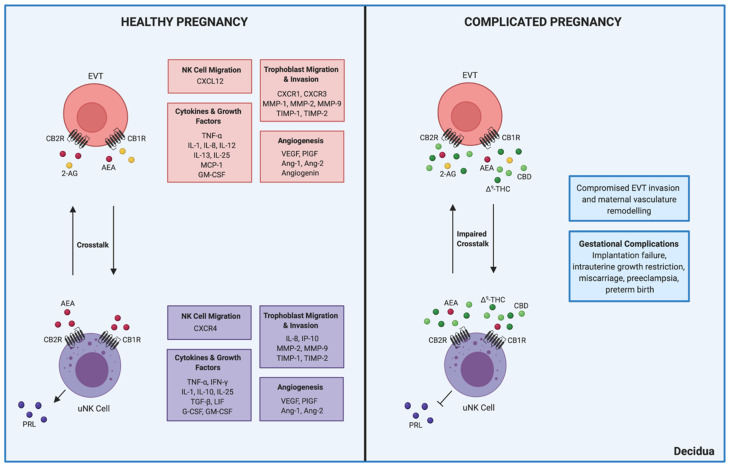
Crosstalk between uNK cells and EVTs in healthy and complicated pregnancies. uNK cells and EVTs express the primary cannabinoid receptors, CB1R and CB2R [43,47]. While uNK cells have only been shown to participate in local AEA production, EVTs can synthesize both AEA and 2-AG [43,47]. In healthy pregnancy, communication between uNK cells and EVTs is necessary for peripheral NK cell migration to the decidua, EVT migration and invasion, regulation of cytokines and growth factors, and angiogenesis [77,78,82,86,87]. However, pregnancy complicated by cannabis use is associated with inhibited intracellular cAMP activity that results in impaired PRL secretion and decidualization [121,122]. Exposure to ∆^9^-THC, and perhaps CBD, can thus alter uNK cell-EVT crosstalk, which may lead to gestational complications through compromised invasion and maternal vasculature remodelling. Unfortunately, the effects of CBD in the decidua and placenta remain undefined in comparison to those of ∆^9^-THC. The long arrows shown in the panels represent interactions between uNK cells and EVTs. In the healthy pregnancy panel, the short arrow signifies the secretion of PRL by uNK cells, which is replaced with an inhibitory arrow in the complicated pregnancy panel. Moreover, the major functions of uNK cells and EVTs in pregnancy are shown in purple and red boxes, respectively, and the overall implications of impaired crosstalk are shown in blue. Created with BioRender.com (accessed on 7 March 2021).

**Figure 2 biomedicines-09-00267-f002:**
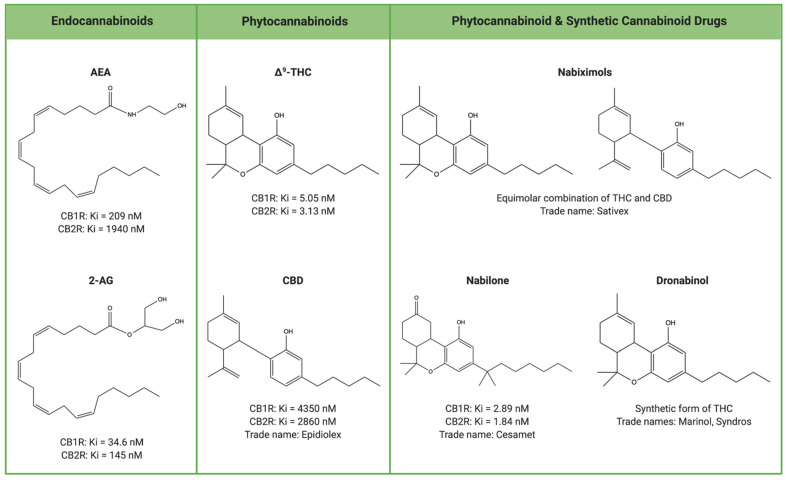
Comparison of cannabinoid structures and receptor binding affinities. Cannabinoids can be classified into three major groups: endocannabinoids, phytocannabinoids, and synthetic cannabinoids. The structures and binding affinities of AEA, 2-AG, Δ^9^-THC, CBD, and nabilone are displayed. While the structures of nabiximols and dronabinol are also shown, there is currently no published information for the Ki values of these compounds. Created with BioRender.com (accessed on 7 March 2021).

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
