# Peer review of "Cannabinoid Signalling in Immune–Reproductive Crosstalk during Human Pregnancy"

_biomedicines, 2021, doi:10.3390/biomedicines9030267_

Round 1
Reviewer 1 Report
In their Review entitled
„Cannabinoid Signalling in Immune-Reproductive Crosstalk During Human Pregnancy“ Harmeet Gurm, Jeremy A. Hirota and Sandeep Raha give information on the implications of ECS-mediated crosstalk between human uNK cells and trophoblasts at the maternal-fetal interface. They focus on the prerequisites for pregnancy success and associated gestational complications and further shortly cover the current status of pharmacotherapy.
Researching in the field of placentology, this reviewer greatly enjoyed reading this nicely balanced review article!
There are minor issues that should be addressed before publication:
While the Figure design is very nice in general,
this reviewer is unhappy with the way things were structured in Figure 1.
This Figure is supposed to show the crosstalk between uNK cells and EVT. However,
when looking at it, it shows 2 compartments decidua/placenta and the displacental crosstalk. Also: The EVT is depicted on the placental side, while it should be placed in the decidual side where the uNK/EVT crosstalk takes place.
Figure Legend 1: please give references for these statements
Revise sentence:
Anandamide (AEA) was the first identified endocanna-binoid in 1992, following which its main biosynthetic
Shouldn’t it read: „after which“
Author Response
We thank the reviewer for their thoughtful feed and implemented all suggested changes.
- Figure 1 has been relabeled as per the reviewers suggestion, with EVT/uNK interactions taking place in the decidual region of the depicted figure.
- Multiple citations have been inserted in figure legend 1 (Lines 318-326) to fully support the summative nature of this figure
-
Line 60: Changed “following which” to “after which”, as per the Reviewer's comment.

Reviewer 2 Report
- The authors have explained some of the basic concepts in ECS. I would recommend adding more concepts in ECS.
- The conclusion needs to be elaborated.
- The review is interesting and adds more information to the ECS research area.
Author Response
We thank the reviewer for their thoughtful comments and have addressed them to the best of our ability.
1. While we recognized that the reviewers comment about about adding more concepts about the ECS is valuable, we were unsure as to the specifics of what the reviewer wished us to insert. We originally limited ourselves to a summary of the ECS as is applied to trophoblasts, the decidua as well as uNK cells. Based on the reviewers feedback we inserted a statement to better justify to the reviewer. and the readers, the rationale for limiting the ECS discussion:
Lines 54-57: Inserted “Unfortunately, research surrounding ECS regulation in peripheral tissues is relatively limited, and has primarily focused on the central nervous system wherein the ECS is implicated in assisting with pain perception and cognitive functions [12].
We hope this will comply with the reviewers request.
2. The Reviewer requested that the conclusion should be elaborated. In order keep the article succinct, we thoughtfully elaborated the conclusion in the following manner:
Lines 544-546: Inserted “Furthermore, components of the ECS are involved in regulating diverse biological processes, including immune and reproductive events in pregnancy”, as per Reviewer 2’s comment on elaborating the conclusion.
Lines 547-550: Inserted “An important consideration is that phytocannabinoids lack the retrograde feedback mechanism utilized by endocannabinoids and can thus continuously activate the cannabinoid receptors. In particular, activation of CB1R by phytocannabinoids…”.
Lines 551-554: Inserted “As a consequence of cannabis exposure, the crosstalk between uNK cells and EVTs can become compromised, resulting in impaired invasion, angiogenesis and potentially, later pregnancy events that promote fetal development”.
Line 555: Inserted “In addition,” to beginning of paragraph 2.
We hope this will meet the reviewers satisfaction and thank for their time in providing the feedback.

Round 2
Reviewer 1 Report
Thank you for implementing my suggestions!
Reviewer 2 Report
No specific comments